# Vulnerability Predictors of Post-Vaccine SARS-CoV-2 Infection and Disease—Empirical Evidence from a Large Population-Based Italian Platform

**DOI:** 10.3390/vaccines10060845

**Published:** 2022-05-26

**Authors:** Giovanni Corrao, Matteo Franchi, Danilo Cereda, Francesco Bortolan, Olivia Leoni, Catia Rosanna Borriello, Petra Giulia Della Valle, Marcello Tirani, Giovanni Pavesi, Antonio Barone, Michele Ercolanoni, Jose Jara, Massimo Galli, Guido Bertolaso

**Affiliations:** 1National Centre for Healthcare Research and Pharmacoepidemiology, University of Milano-Bicocca, 20126 Milan, Italy; giovanni.corrao@unimib.it; 2Unit of Biostatistics, Epidemiology and Public Health, Department of Statistics and Quantitative Methods, University of Milano-Bicocca, 20126 Milan, Italy; 3Directorate General for Health, Lombardy Region, 20124 Milan, Italy; danilo_cereda@regione.lombardia.it (D.C.); francesco_bortolan@regione.lombardia.it (F.B.); olivia_leoni@regione.lombardia.it (O.L.); catia_rosanna_borriello@regione.lombardia.it (C.R.B.); dellavalle_petra_stg@regione.lombardia.it (P.G.D.V.); marcello_tirani@regione.lombardia.it (M.T.); giovanni_pavesi@regione.lombardia.it (G.P.); 4Azienda Regionale per l’Innovazione e gli Acquisti (ARIA) S.p.A., 20124 Milan, Italy; antonio.barone@ariaspa.it (A.B.); michele.ercolanoni@ariaspa.it (M.E.); jose.jara@ext.ariaspa.it (J.J.); 5Infectious Diseases Unit, Luigi Sacco Hospital, 20157 Milan, Italy; massimo.galli@unimi.it; 6Department of Biomedical and Clinical Sciences, University of Milan, 20157 Milan, Italy; 7Vaccination Campaign Management, Lombardy Region, 20124 Milan, Italy; bertolaso1@gmail.com

**Keywords:** SARS-CoV-2, COVID-19, vaccines, vulnerability, predictors

## Abstract

We aimed to identify individual features associated with increased risk of post-vaccine SARS-CoV-2 infection and severe COVID-19 illness. We performed a nested case–control study based on 5,350,295 citizens from Lombardy, Italy, aged ≥ 12 years who received a complete anti-COVID-19 vaccination from 17 January 2021 to 31 July 2021, and followed from 14 days after vaccine completion to 11 November 2021. Overall, 17,996 infections and 3023 severe illness cases occurred. For each case, controls were 1:1 (infection cases) or 1:10 (severe illness cases) matched for municipality of residence and date of vaccination completion. The association between selected predictors (sex, age, previous occurrence of SARS-CoV-2 infection, type of vaccine received, number of previous contacts with the Regional Health Service (RHS), and the presence of 59 diseases) and outcomes was assessed by using multivariable conditional logistic regression models. Sex, age, previous SARS-CoV-2 infection, type of vaccine and number of contacts with the RHS were associated with the risk of infection and severe illness. Moreover, higher odds of infection and severe illness were significantly associated with 14 and 34 diseases, respectively, among those investigated. These results can be helpful to clinicians and policy makers for prioritizing interventions.

## 1. Introduction

The emergence of novel SARS-CoV-2 variants [1] and the decreasing trend in the titers of antibodies in vaccinated individuals [2] have raised public health concerns regarding the efficacy and duration of protection induced by first-generation vaccines [3]. The persistence of neutralizing antibodies and the degree of protection they confer remain largely unknown. Hence, understanding the risk of both infection with SARS-CoV-2 and severe clinical manifestations of COVID-19 after vaccination is completed provides an avenue to assess the path to protection against COVID-19 [4]. Finally, identifying the predictors of post-vaccine SARS-CoV-2 infection and COVID-19 disease is mandatory for prioritizing interventions.

With the aim to shed light on this field, we leveraged the integrated platform of the vaccination campaign of Lombardy, the largest Italian region, including almost nine million candidates for vaccination (i.e., beneficiaries of the Regional Health Service (RHS) aged 12 years or older). Using this observational database, we explored demographic and clinical factors associated with both increased risk of post-vaccine SARS-CoV-2 infection and severe COVID-19 illness.

## 2. Materials and Methods

The target population includes 5,351,085 beneficiaries of the Regional Health Service aged 12 years or older, who completed a vaccination program (i.e., two doses of vaccine provided by Pfizer, Moderna or Oxford-AstraZeneca, or one dose of vaccine by Janssen) from 17 January to 31 July 2021. The 790 citizens who experienced SARS-CoV-2 infection and/or COVID-19 hospital admission or death within 14 days after vaccine completion were excluded. The remaining 5,350,295 citizens entered the study cohort and were followed from 14 days after vaccine completion (under the assumption that immune coverage is achieved 2 weeks after receiving the vaccine [5]) to outcome occurrence (see after), death for a cause different from COVID-19, emigration, or 11 November 2021, whichever occurred earliest (follow-up).

A population-based platform was realized since starting vaccination campaign by means of record linking (i) the COVID-19 vaccination registry (collecting date, type, and dose of vaccine dispensed), (ii) the registry of confirmed diagnosis of SARS-CoV-2 infection (collecting ascertained infections and hospital admissions, emergency-room access and deaths due to COVID-19), and (iii) the health care utilization database (collecting various types of information, including causes of death, inpatient diagnoses supplied by public or private hospitals, and outpatient drug dispensation). All these different data may be interconnected through a deterministic record linkage because a single individual identification code is used in all databases. To preserve privacy, each identification code was deidentified automatically, with this inverse process being allowed only for the RHS on request from judicial authorities. Further details of the health care databases used in the context of COVID-19 in Lombardy have been reported [6].

A nested case–control design was adopted by separately listing cases who during follow-up experienced the first occurrence of either (i) infection documented by nasopharyngeal swab testing positive for SARS-CoV-2 via a PCR test in any clinical setting regardless of the presence of symptoms; or (ii) COVID-19 hospital admission, including admission to an intensive care unit, or death. These were denoted infection and severe illness cases, respectively. The date of outcome occurrence was denoted the index date. For each case patient, one (infection cases) or ten (severe illness cases) controls were randomly selected from the study cohort who had not experienced the outcomes at the index date, to be matched for municipality of residence and date of vaccination completion.

The following information were retrieved at the individual level: gender, age at cohort entry, previous occurrence of infection from SARS-CoV-2, type of vaccine received (categorized as mRNA-based and adenovirus-vectored vaccines), and medical pathway traced from contacts with the RHS during 2018 and 2019. The latter comprised categories of the number of contacts with the RHS, and the presence/absence of 59 diseases/conditions (candidate predictors) traced though hospital admissions and drug prescriptions. The list of candidate predictors included practically all nosologic categories and was prepared taking into consideration morbidity and mortality predictors reported by selected systematic reviews and meta-analyses [7,8,9], as well as in a population-based cohort study [10]. The list of candidate predictors, and the corresponding codes, are reported in Appendix A.

With the aim of investigating the strength of association between the above reported factors and the odds of experiencing the outcome of interest, conditional logistic regression was fitted by including all these covariates in a unique model. With the aim of flexibly modelling, the dose–response relationship between age and the odds of the considered outcomes, restricted cubic splines were used with four knots, and the results were presented as a nonlinear trend in odds ratio, with 95% confidence bands, using 40 years old as reference age [11]. Subsequently, with the aim of investigating the association between each candidate predictor and the outcome of interest, conditional logistic regression was fitted by including one condition (which is entered in the model as a dichotomous variable, with a value of 1 or 0 according to whether the specific condition was or was not recorded at least during the years 2018 and 2019) while adjusting for the above considered covariates. Only conditions affecting at least 10 cases of either infection or severe illness were included in this analysis.

## 3. Results

Among the 5,350,295 citizens included into the study cohort, 46.9% were men, their mean age was 57.7 years (SD 18.0 years) and 76.9% of them received an mRNA-based vaccine. Cohort members accumulated 24,849,267 person-months of observation (on average almost 4.6 months for each of them) and generated 17,996 infections and 3023 severe illnesses (incidence rates being 7.2 and 1.2 cases per 10,000 person-months, respectively). The 17,996 infection cases were matched with as many controls, while the 3023 severe illness cases were matched with 30,230 controls.

Figure 1 shows that the relationship between age and considered outcomes had opposite patterns. The odds of SARS-CoV-2 infection reached the highest peak at the age of 30 years, followed by decreasing values until 60 years and relatively stable ones afterwards. Conversely, the lowest values in odds of severe COVID-19 were reached around the age of 20 years, followed by increasing values which reached the highest peak around the age of 95 years.

Other risk factors of both post-vaccine SARS-CoV-2 infection and severe COVID-19 illness are pictured in Figure 2. Male gender significantly increased the odds of severe illness. A trend towards increasing odds of both infection and severe illness was observed as the number of contacts with the RHS increased. Having had a previous SARS-CoV-2 infection was a significant protective factor against both post-vaccine infection and severe illness. Having been vaccinated with an mRNA-based product was a protective factor against infection.

Figure 3 shows the association strengths between 49 candidate predictors which affected at least 10 cases of infection. Significant higher odds of infection were associated with 14 diseases/conditions, i.e., the 29% of the investigated factors. The most associated disease/conditions were chronic kidney disease (OR = 1.80, 95% CI 1.38 to 2.35), dementia/Alzheimer’s disease (OR = 1.62, 1.27 to 2.05), and transplantation (OR = 1.48, 1.05 to 2.08).

Similarly, Figure 4 shows the associations between 43 candidate predictors which affected at least 10 cases of severe illness. Among these, 34 significantly increased the odds of severe illness, i.e., the 79% of the investigated factors. The most associated disease/conditions were chronic kidney disease (OR = 2.95, 95% CI 2.32 to 3.76), acute respiratory infections (OR = 2.84, 1.80 to 4.47), and other mental disorders (OR = 2.53, 1.59 to 4.04).

## 4. Discussion

The current study based on real-world data from more than 5 million people who completed vaccination against COVID-19 identified an extensive set of factors increasing the risk of post-vaccine SARS-CoV-2 infection and/or severe COVID-19 illness. Factors such as age younger than 40 years, frequent contacts with the RHS, absence of previous ascertained infection, having received an adenovirus-vectored vaccine, and several conditions including gout, anemias, mental disorders, organ transplantation, chronic cardiovascular, respiratory and kidney diseases, and disease of the skin and musculoskeletal systems, significantly increased the risk of SARS-CoV-2 infection. Factors such as old age, male gender, frequent contacts with the RHS, and diseases/conditions affecting practically almost all the organs and systems significantly increased the risk of severe COVID-19 illness. This may provide a useful reference for establishing priority in the booster vaccination programs, as well as for access to future treatment options, such as monoclonal antibodies.

Our study provides the following additional results. One, the opposite pattern of the dose–response relationship between age and the risk of SARS-CoV-2 infection and/or severe COVID-19 illness likely depends on the higher occasion of contagion for younger citizens, as well as the higher frailty of older citizens, making them on average more vulnerable to the development of the severe and fatal clinical manifestations of the COVID-19 infection [12]. Two, the observed higher risk of severe illness among males is another widely expected finding. Consistently, it has been reported that despite the number and age of males and females with SARS-CoV-2 infection being comparable, males tend to display more severe disease [13,14,15]. Although several factors have been speculated to account for the disparity, including differences in biology, behavior, occupation, and immune response [16,17], the underlying mechanisms are still unclear [13]. Three, having received an adenovirus-vectored vaccine offered lower protection against the SARS-CoV-2 infection than having received an mRNA-based vaccine. Although our study does not provide information on underlying mechanisms, differential patterns of immunogenicity from available vaccine platforms have been reported, with antibody responses being 2.9-fold higher following the mRNA-based vaccine than the adenovirus-vectored vaccine [18]. Four, our study provides further evidence that the risk of post-vaccine SARS-CoV-2 infection is strongly reduced among individuals who already experienced SARS-CoV-2 infection than naïve individuals. With respect to the available evidence, however, our study goes beyond the comparison of immune response in recovered and naïve individuals [19,20,21,22,23,24,25,26], extending such evidence to the protective action of previous infection against the post-vaccine reinfection [27]. Finally, several conditions and diseases of which vaccinated citizens suffered strongly affected their risk of post-vaccine infection and illness. It is interesting that while not even a one-third of the investigated diseases affected the risk of infection, almost four in five of them, belonging to all systems and organs, were strongly involved in the risk of severe disease. This confirms the now established notion that alterations in the structure and function of virtually all organs and systems of the body may adversely affect resistance to the COVID-19 disease [10].

The present study had several points of strength. One, this study provides the largest and most robust available evidence of the post-vaccine risk factors of infection of COVID-19 and its clinical consequences. Two, this study was based on a very large population and included all ages that were regarded as suitable for the COVID-19 vaccination. This allowed a large accumulation of person-months, which means that although post-vaccine infections and cases of severe illness are very rare events, this study was sufficiently powered to address its primary goal.

Limitations are that the predictors of COVID-19 we searched for are restricted to those routinely collected and available in the administrative databases, i.e., hospital admissions and drugs dispensed. However, additional predisposing factors may impact the risk of COVID-19 infection and severe illness. For example, a recent study conducted in Italy supported a role of the ABO blood type in the development of symptomatic disease with a higher risk in subjects with blood type A and a protective effect of blood types B and O [28]. In addition, our system for tracking diseases did not capture the severity of associated comorbidities. Furthermore, health services and treatments supplied by private providers were not captured by our analysis. Moreover, misdiagnosis (due to poor accuracy in reporting diagnoses and comorbidities) and upcoding in hospital records (sometimes in pursuit of higher reimbursements) might have underestimated the prevalence of patients affected by the considered conditions. Finally, outcome misclassification may have affected this study, because not all citizens in whom the infection occurred were tracked and some patients with severe symptoms might have been treated at home.

## 5. Conclusions

In conclusion, by a large population-based platform accomplished for monitoring trend and impact of vaccine campaign in the largest Italian region, we identified an extensive set of factors increasing the risk of SARS-CoV-2 infection and/or severe COVID-19 illness. Factors such as age younger than 40 years, frequent contacts with the RHS, absence of previous ascertained infection, having received an adenovirus-vectored vaccine, and several conditions including gout, anemias, mental disorders, organ transplantation, chronic cardiovascular, respiratory and kidney diseases, and disease of the skin and musculoskeletal systems, significantly increased the risk of SARS-CoV-2 infection. Factors such as elderly, male gender, frequent contacts with the RHS, and diseases/conditions belonging to all the considered nosologic districts, strongly increased the risk of severe COVID-19 illness. This suggests that post-vaccine vulnerability to severe clinical manifestations of SARS-CoV-2 infection may be mainly affected by clinical frailty, possibly due to comorbidities, rather (other) than to specific disorders. Our study confirms that both SARS-CoV-2 infections and severe COVID-19 illness may occur after the vaccine cycle is completed, although with low incidence. These findings would support efforts to maximize both vaccine uptake with two doses and fulfilment with individual protection measures. Clinicians and policy makers can use our results for prioritizing interventions, while researchers can utilize our findings to develop prognostic models that could eventually facilitate decision making.

## Figures and Tables

**Figure 1 vaccines-10-00845-f001:**
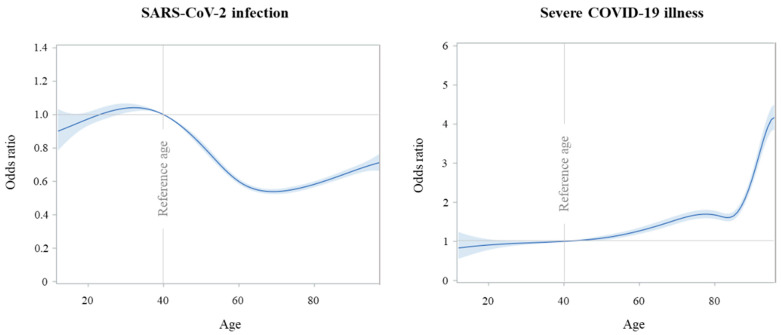
Flexibly modelling the relationship between age at vaccine completion and the odds of post-vaccine SARS-CoV-2 infection (**left panel**) and severe COVID-19 illness (**right panel**).

**Figure 2 vaccines-10-00845-f002:**
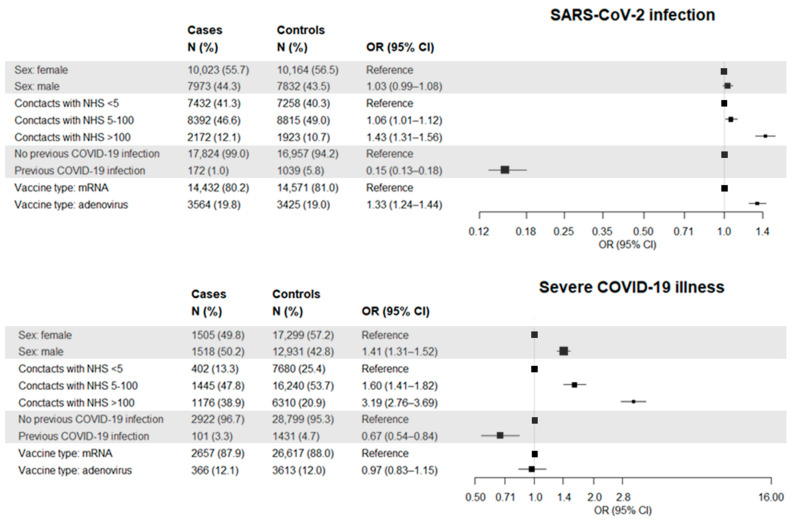
Forest plots showing the association between selected features of the study cohort (citizens who completed scheduled vaccine plan) and the odds of post-vaccine SARS-CoV-2 infection (**top panel**) and severe COVID-19 illness (**bottom panel**). Squares represents the point estimates (i.e., the odds ratios) and the straight line represents the 95% confidence interval.

**Figure 3 vaccines-10-00845-f003:**
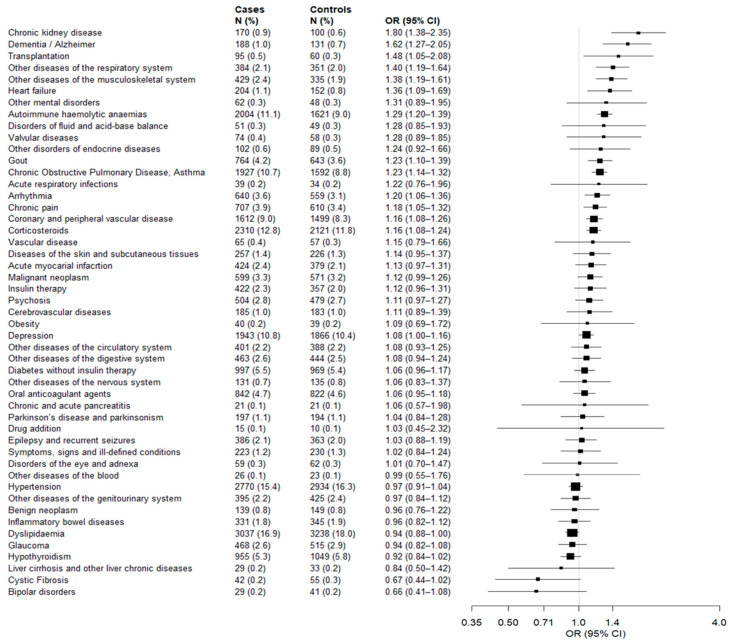
Forest plots showing the association between 49 diseases/conditions members of the study cohort (citizens who completed scheduled vaccine plan) suffered from and the odds of post-vaccine SARS-CoV-2 infection. Odds ratios were estimated by using conditional logistic regressions, by including one condition at a time, while adjusting for age, sex, number of contacts with the Regional Health Service, previous COVID-19 infection and vaccine type. The 49 diseases/conditions are sorted based on decreasing values of the odds ratio. Squares represents the point estimates (i.e., the odds ratios) and the straight line represents the 95% confidence interval.

**Figure 4 vaccines-10-00845-f004:**
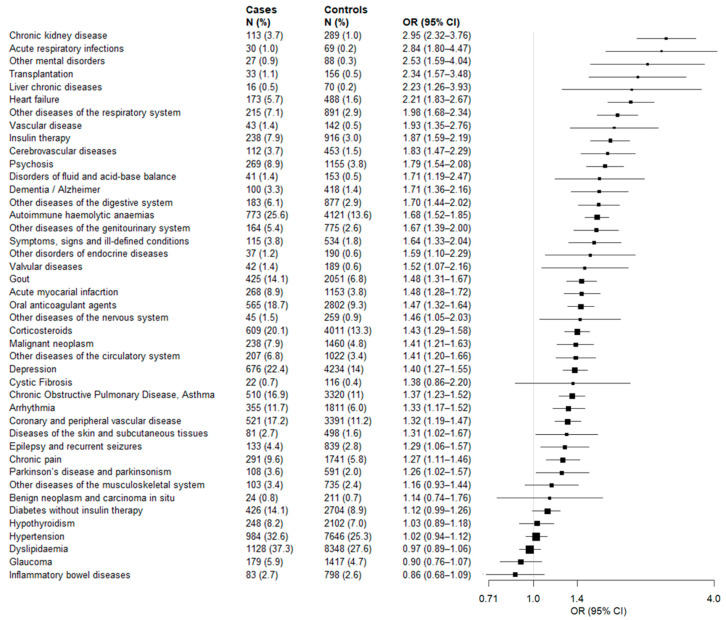
Forest plots showing the association between 43 diseases/conditions members of the study cohort (citizens who completed scheduled vaccine plan) suffered from and the odds of severe COVID-19 illness. Odds ratios were estimated by using conditional logistic regressions, by including one condition at a time, while adjusting for age, sex, number of contacts with the Regional Health Service, previous COVID-19 infection and vaccine type. The 43 diseases/conditions are sorted based on decreasing values of the odds ratio. Squares represents the point estimates (i.e., the odds ratios) and the straight line represents the 95% confidence interval.

## Data Availability

The data that support the findings of this study are available from the Lombardy region, but restrictions apply to the availability of these data, which were used under license for the current study, and so are not publicly available. Data are, however, available from the Lombardy region upon reasonable request.

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
