# Peer review of "Vulnerability Predictors of Post-Vaccine SARS-CoV-2 Infection and Disease—Empirical Evidence from a Large Population-Based Italian Platform"

_vaccines, 2022, doi:10.3390/vaccines10060845_

Round 1

Reviewer 1 Report

The manuscript by Corrao et al., describes the findings of a large population study summarizing the risk factors of SARS-CoV-2 infection and severe outcomes following two-dose vaccination. 

The study is straightforward,  results are well presented and discussed. The results of the study are of importance to researchers and also policymakers.

Author Response

The manuscript by Corrao et al., describes the findings of a large population study summarizing the risk factors of SARS-CoV-2 infection and severe outcomes following two-dose vaccination.

The study is straightforward, results are well presented and discussed. The results of the study are of importance to researchers and also policymakers.

Re: We thank the reviewer for his/her positive comments on our manuscript.

Reviewer 2 Report

A very nice manuscript about the risk factors associated with covID-19 infections after 2 doses of vaccine.

In the footers below the forest plots (Figures 3 & 4), please add a comment that the effect measures are adjusted for sex, vaccine type, etc. Were they adjusted for age as well? Please clarify.

Author Response

A very nice manuscript about the risk factors associated with covID-19 infections after 2 doses of vaccine.

In the footers below the forest plots (Figures 3 & 4), please add a comment that the effect measures are adjusted for sex, vaccine type, etc. Were they adjusted for age as well? Please clarify.

Re: We thank the reviewer for his/her positive comments on our manuscript. As suggested, we clearly specified in the footers of Figures 3 and 4 that “Odds ratios were estimated by using conditional logistic regressions, by including one condition at a time, while adjusting for age, sex, number of contacts with the NHS, previous COVID-19 infection and vaccine type”.

Reviewer 3 Report

This work is to describe " Vulnerability predictors of post-vaccine SARS-CoV-2 infection 2 and disease. Empirical evidence from a large population-based 3 Italian platform". It is very interesting and well-structured. However, there are some parts should be revised.

Minor Revision

The English form should be revised.

Also, the Authors should be carefully check all the acronyms present in the text (i.e. SARS-CoV-2, COVID-19,…).

The Material and Methods paragraph has some missing points. Please describe the inclusion/exclusion criteria in the study setting.

Also, in Material and Methods, it should be clarified the number of dropouts, in lines 63-65, as well as the platform used for the record linking. It should be added the systematic review tool used and the metanalysis software.

The results are clear and support their conclusion, however, the figures’ description should be implemented in the text and an objective explanation is needed.

In line 140, it is written “Significant higher odds of…”, how significant? Which statistic method was used?

There are some write-typing errors. Please check carefully lines 181 (2·9- times and 226 (10).

Author Response

Reviewer 3

This work is to describe " Vulnerability predictors of post-vaccine SARS-CoV-2 infection 2 and disease. Empirical evidence from a large population-based 3 Italian platform". It is very interesting and well-structured.

Re: We thank the Reviewer for their positive comment and careful review, which helped improve the manuscript.

However, there are some parts should be revised.

Minor Revision

The English form should be revised.

Re: We carefully checked and corrected English language in the whole manuscript.

Also, the Authors should be carefully check all the acronyms present in the text (i.e. SARS-CoV-2, COVID-19,…).

Re: We checked and corrected all the acronyms in the whole manuscript.

The Material and Methods paragraph has some missing points. Please describe the inclusion/exclusion criteria in the study setting.

Re: We rephrased the first paragraph of the Material and Methods (lines 58-68), making inclusion and exclusion criteria more clear.

Also, in Material and Methods, it should be clarified the number of dropouts, in lines 63-65, as well as the platform used for the record linking. It should be added the systematic review tool used and the metanalysis software.

Re: In the first paragraph of the Material and Methods (lines 62-64), we now clearly specified that “the 790 citizens who experienced SARS-CoV-2 infection and/or COVID-19 hospital admission or death within 14 days after vaccine completion, were excluded”.

We specified in the second paragraph of the Material and Methods (lines 75-77) that “all these different data may be interconnected through a deterministic record-linkage because a single individual identification code is used in all databases”.

We did not use any tool for systematic review and meta-analyses, since our study is not a meta-analysis. We used forest-plot (that is a graphical representation commonly used for reporting results of meta-analyses) only for graphically show the results of our analyses, but we did not applied methods of meta-analysis (as stated at line 81, our study is a nested case-control study).

The results are clear and support their conclusion, however, the figures’ description should be implemented in the text and an objective explanation is needed.

Re: As suggested by the reviewer, we added a description of Figures 3 and 4 in the Results section, as follows (lines 143-154): “Figure 3 shows the association strengths between 49 candidate predictors which affected at least 10 cases of infection. Significant higher odds of infection were associated with 14 diseases/conditions, i.e., the 29% of the investigated factors. The most associated disease/conditions were chronic kidney disease (OR=1.80, 95% CI 1.38 to 2.35), dementia/Alzheimer (OR=1.62, 1.27 to 2.05), and transplantation (OR=1.48, 1.05 to 2.08).

Similarly, Figure 4 shows the associations between 43 candidate predictors which affected at least 10 cases of severe illness. Among these, 34 significantly increased the odds of severe illness, i.e., the 79% of the investigated factors. The most associated disease/conditions were chronic kidney disease (OR=2.95, 95% CI 2.32 to 3.76), acute respiratory infections (OR=2.84, 1.80 to 4.47), and other mental disorders (OR=2.53, 1.59 to 4.04).”.

In line 140, it is written “Significant higher odds of…”, how significant? Which statistic method was used?

Re: The statistical significance of the odds ratio was based on the 95% confidence intervals. Indeed, a lower bound of the confidence intervals higher than 1 implies that the odds in infection (or of severe illness) is statistically higher in individuals presenting that disease/conditions, as compared to those who did not suffer from it, at a significance level of 0.05.

There are some write-typing errors. Please check carefully lines 181 (2·9- times and 226 (10).

Re: We thank the reviewer. We checked the typos.

Reviewer 4 Report

The manuscript entitled “Vulnerability predictors of post-vaccine SARS-CoV-2 infection and disease. Empirical evidence from a large population-based Italian platform” aims to explore the vulnerability predictors of SARS-CoV-2 infection and COVID-19 illness in subjects who received a complete vaccination. The authors performed a nested case-control study on a population of more than 5 million citizens who received vaccination for the purpose to identify demographic and clinical features associated with increased risk of SARS-CoV-2 infection and severe disease.

The strength of the study is undoubtedly represented by the large studied population, although information on the socio-demographic and clinical characteristics of the subjects, and the severity of comorbidities are lacking. Additional predisposing factors for infection and severe illness have been evaluated in other studies, such as blood group and other genetic factors. Furthermore, details on the SARS-CoV-2 variants (alpha? Delta?) and the time elapsed between the last vaccination dose and the occurrence of SARS-CoV-2 infection could be important.

In the Introduction section, some literature references on the predisposing factors to infection and severe disease also in unvaccinated subjects could be included and, the data could be compared with the results of the study.

In the Abstract section (page 1, line 25) the authors indicate the date of December 27, 2020, as the start date of the study, while in the materials and methods section (page 2, line 56) they refer to January 17 as the start date. Please, correct it or explain the different dates.

In the Abstract section, lines 32-33, the sentence is unclear. Please, revise it.

On page 1, line 41, please, correct “has” with “have”.

The authors identify the predisposing factors to the increased risk of infection and severe COVID-19 illness. “Infection” is documented by nasopharyngeal swab positive for SARS-CoV-2 via a PCR test, while “severe illness cases” include all subjects admitted to hospital for COVID-19, in the intensive care unit, or death. Subjects with mild or moderate disease are often admitted to the hospital because of advanced age or suffering from underlying comorbidities.

On page 2, line 95 the authors report that morbidity and mortality predictors were prepared by selected systematic reviews and meta-analyses. Bibliographic references include also a population-based cohort study. Please, specify it in the test.

On page 3, line 121 "was" is incorrect. Please, correct it.

On page 3, line 132 please replace "factors" with "factor"

In the Results section, the authors report the number and percentage of underlying conditions/diseases associated with infection and severe disease. The most significant factors or significantly associated with one group and not the other could be mentioned in the text.

Numerous researchers have reported additional predisposing factors for COVID-19 infection or severe illness, such as genetic factors, ABO blood system, etc. Considering the large population available to the authors, they could cite in the discussion some research that has already investigated the "role of ABO in COVID-19 patients" in their country and propose it as a further topic of investigation.

Author Response

The manuscript entitled “Vulnerability predictors of post-vaccine SARS-CoV-2 infection and disease. Empirical evidence from a large population-based Italian platform” aims to explore the vulnerability predictors of SARS-CoV-2 infection and COVID-19 illness in subjects who received a complete vaccination. The authors performed a nested case-control study on a population of more than 5 million citizens who received vaccination for the purpose to identify demographic and clinical features associated with increased risk of SARS-CoV-2 infection and severe disease.

The strength of the study is undoubtedly represented by the large studied population, although information on the socio-demographic and clinical characteristics of the subjects, and the severity of comorbidities are lacking. Additional predisposing factors for infection and severe illness have been evaluated in other studies, such as blood group and other genetic factors. Furthermore, details on the SARS-CoV-2 variants (alpha? Delta?) and the time elapsed between the last vaccination dose and the occurrence of SARS-CoV-2 infection could be important.

Re: We thank the reviewer for his/her positive comments on our manuscript. We agree that, other the ones investigated, additional predisposing factors may be associated to the risk of infection and severe illness. However, as stated in the Discussion (lines 207-209), “predictors of COVID-19 we searched for are restricted to those routinely collected and available in the administrative databases”, thus precluding the possibility of evaluating other factors. Moreover, we agree that time elapsed between the last vaccination dose and the occurrence of SARS-CoV-2 infection could be important. For this reason, our study design took into account this aspect, by matching cases and controls by the date of vaccination completion. In this way, the time elapsed between the last vaccination dose and the occurrence of SARS-CoV-2 infection (or sever illness) was the same for the case and the corresponding controls.

In the Introduction section, some literature references on the predisposing factors to infection and severe disease also in unvaccinated subjects could be included and, the data could be compared with the results of the study.

Re: We agree with the reviewer’s suggestion of comparing post-vaccine risk factors with risk factors in unvaccinated individuals. At this purpose, in the Discussion (lines 189-192) we stated that “the observed higher risk of severe illness among males is another widely expected finding. Consistently, it has been reported that despite the number and age of males and females with SARS-CoV-2 infection are comparable, males tend to display more severe disease [13-15]”. Moreover (lines 205-211), “several conditions and diseases of which vaccinated citizens suffered strongly affected their risk of post-vaccine infection and illness… This confirms the now established notion that alterations of the structure and function of virtually all organs and systems of the body may adversely affect resistance to the COVID-19 disease [10]”. In order to not weigh down the Introduction section, as well as to not confound the reader on the main objective of the study, we prefer not to add this point in the Introduction, but only to point it out in the Discussion.

In the Abstract section (page 1, line 25) the authors indicate the date of December 27, 2020, as the start date of the study, while in the materials and methods section (page 2, line 56) they refer to January 17 as the start date. Please, correct it or explain the different dates.

Re: We corrected the starting date in the Abstract (January, 17 2021).

In the Abstract section, lines 32-33, the sentence is unclear. Please, revise it.

Re: We rephrased the sentence in the Abstract, as follows: “Sex, age, previous SARS-CoV-2 infection, type of vaccine and number of contacts with the RHS were associated with the risk of infection and severe illness” (lines 33-35).

On page 1, line 41, please, correct “has” with “have”.

Re: We corrected the verbal form, as suggested by the reviewer.

The authors identify the predisposing factors to the increased risk of infection and severe COVID-19 illness. “Infection” is documented by nasopharyngeal swab positive for SARS-CoV-2 via a PCR test, while “severe illness cases” include all subjects admitted to hospital for COVID-19, in the intensive care unit, or death. Subjects with mild or moderate disease are often admitted to the hospital because of advanced age or suffering from underlying comorbidities.

Re: We agree with the reviewer that elderly patients, as well as those presenting comorbidities, have a greater probability of being admitted to hospital. For this reason, the association between selected clinical or non-clinical factors and the risk of infection or severe disease were adjusted by age, as well as by the number of previous contacts with the Regional Health Service in the years 2018-2019 (that is a proxy of the “clinical condition” of each individual).

On page 2, line 95 the authors report that morbidity and mortality predictors were prepared by selected systematic reviews and meta-analyses. Bibliographic references include also a population-based cohort study. Please, specify it in the test.

Re: We modified the sentence as follows (lines 96-99): “The list of candidate predictors included practically all nosologic categories and was pre-pared taking into consideration morbidity and mortality predictors reported by selected systematic reviews and meta-analyses [7-9], as well as in a population-based cohort study [10]”.

On page 3, line 121 "was" is incorrect. Please, correct it.

Re: We replaced “was” with “were”.

On page 3, line 132 please replace "factors" with "factor"

Re: We replaced “factors” with “factor”.

In the Results section, the authors report the number and percentage of underlying conditions/diseases associated with infection and severe disease. The most significant factors or significantly associated with one group and not the other could be mentioned in the text.

Re: As suggested by the reviewer, we further described Figures 3 and 4 in the Results section, as follows (lines 143-154): “Figure 3 shows the association strengths between 49 candidate predictors which affected at least 10 cases of infection. Significant higher odds of infection were associated with 14 diseases/conditions, i.e., the 29% of the investigated factors. The most associated disease/conditions were chronic kidney disease (OR=1.80, 95% CI 1.38 to 2.35), dementia/Alzheimer (OR=1.62, 1.27 to 2.05), and transplantation (OR=1.48, 1.05 to 2.08).

Similarly, Figure 4 shows the associations between 43 candidate predictors which affected at least 10 cases of severe illness. Among these, 34 significantly increased the odds of severe illness, i.e., the 79% of the investigated factors. The most associated disease/conditions were chronic kidney disease (OR=2.95, 95% CI 2.32 to 3.76), acute respiratory infections (OR=2.84, 1.80 to 4.47), and other mental disorders (OR=2.53, 1.59 to 4.04).”.

Numerous researchers have reported additional predisposing factors for COVID-19 infection or severe illness, such as genetic factors, ABO blood system, etc. Considering the large population available to the authors, they could cite in the discussion some research that has already investigated the "role of ABO in COVID-19 patients" in their country and propose it as a further topic of investigation.

Re: As suggested, we added the following sentence in the Discussion section (lines 221-225): “However, additional predisposing factors may impact the risk of COVID-19 infection and severe illness. For example, a recent study conducted in Italy supported a role of ABO blood type in the development of symptomatic disease with a higher risk in subjects with blood type A and a protective effect of blood types B and O [28].”.